Integrin signaling in tumor biology: mechanisms of intercellular crosstalk and emerging targeted therapies

Li Yifan 1
Peng Shantong 1
Xu Jiatong 1
Liu Wenjie 2
Luo Qi jxnculq2018@163.com 3
1 Queen Mary School, Jiangxi Medical College, Nanchang University , Nanchang , Jiangxi , China
2 The First Clinical College, Jiangxi Medical College, Nanchang University , Nanchang , Jiangxi , China
3 College of Basic Medical Sciences, Nanchang University , Nanchang , Jiangxi , China
Haraguchi Tokuko
Electronic publication date: 2025 May 7
Publication date: 2025
Volume: 13
Electronic Location ID: e19328
Received 2024 Dec 11; Accepted 2025 Mar 25
Copyright: ©2025 Li et al.
Copyright year: 2025
Copyright holder: Li et al.
License: This is an open access article distributed under the terms of the Creative Commons Attribution License, which permits unrestricted use, distribution, reproduction and adaptation in any medium and for any purpose provided that it is properly attributed. For attribution, the original author(s), title, publication source (PeerJ) and either DOI or URL of the article must be cited.
License URL: https://creativecommons.org/licenses/by/4.0/

Keywords: Integrins, Tumor, Targeted therapy

Funding: The authors received no funding for this work.

==============================
Integrins, a family of transmembrane cell adhesion receptors, mediate intercellular and cell–extracellular matrix crosstalk via outside-in and inside-out signaling pathways. Integrins, categorized into 24 distinct combinations of α and β subunits, exhibit tissue-specific expression and perform unique or overlapping roles in physiological and pathophysiological processes. These roles encompass embryonic angiogenesis, tissue repair, and the modulation of tumor cell angiogenesis, progression, invasion, and metastasis. Notably, integrins are significant contributors to tumor development, offering valuable insights into the potential of integrin-targeted diagnostics and therapeutics. Currently, there are various preclinical and clinical trials aiming to harness integrin antagonists that are safe, efficacious, and exhibit low toxicity. Owing to the functional redundancy across integrin types and the complexity of the mechanisms of integrin-mediated multiple key processes associated with tumor biology, challenges exist that impede advancements in integrin-targeted therapy. Nevertheless, innovative strategies focused on integrin modulation represent significant breakthroughs for improving patient care and promoting comprehensive insights into the underlying mechanisms of tumor biology. This review elucidates the impact of integrins on three distinct cell types in multiple key processes associated with tumor biology and explores the emerging integrin-targeted therapeutic approaches for the treatment of tumors, which will provide ideas for optimal therapeutic approaches in the future.

Introduction

Integrins, a class of heterodimeric transmembrane glycoprotein adhesion receptors, composed of α and β subunits and facilitate intercellular and cell–extracellular matrix (ECM) crosstalk, and thus modulate numerous signaling pathways implicated in physiological and pathophysiological conditions (Hamidi & Ivaska, 2018; Takada, Ye & Simon, 2007). There are 18 α and 8 β subunits in humans, forming a total of 24 distinct types of heterodimeric integrins (Fig. 1). Each type of integrin variant exhibits unique expression patterns and exerts specific functions across diverse tissues because their activation depends on distinct ligands that initiate disparate downstream signaling cascades (Takada, Ye & Simon, 2007). Consequently, integrins are categorized into four groups based on their ligand specificity: RGD (Arg-Gly-Asp) receptors, laminin receptors, collagen receptors, and leukocyte-specific receptors.

Figure 1 Twenty-four types of integrins have been identified so far, comprising 18 α and 8 β subunits.

The types of receptors can be subdivided into four groups, including RGD receptors, laminin receptors, collagen receptors, and leukocyte-specific receptors. RGD: Arg-Gly-Asp.

Integrins are distinct from other receptors because they can transmit biological signals bidirectionally through two distinct mechanisms known as “outside-in” and “inside-out” signaling pathways. In the inside-out pathway, intracellular components such as talin and kindlin bind to the cytoplasmic tail of integrins, acting as integrin promoters or inhibitors, and regulate the sensitivity of integrins to their extracellular ligands. In the outside-in signaling pathways, extracellular components bind to integrins and initiate downstream signaling cascades (Lietha & Izard, 2020). Upon binding of extracellular components, integrins aggregate into clusters and communicate with cytoskeletal complexes that further promote integrin clustering, thereby establishing an ECM-integrin-cytoskeleton axis (Giancotti & Ruoslahti, 1999). In addition, the activated integrins can modulate diverse cellular activities by interacting with and activating integral components of cellular signaling pathways. For example, integrins can activate various protein tyrosine kinases including focal adhesion kinases (FAK), Src-family kinases (Src), phosphoinositide-3-kinase (PI3K), and Akt kinases, subsequently mediating angiogenesis and fibroblast migration (Ellert-Miklaszewska et al., 2020; Giancotti & Ruoslahti, 1999).

Integrins play crucial roles in mediating multiple physiological and pathophysiological activities such as in tumors. In particular, previous studies showed that integrins communicating with the tumor microenvironment are invaluable for tumor progression, angiogenesis, lymph angiogenesis, migration, invasion, and metastasis. Indeed, integrins αvβ3, αvβ5, and α5β1 on endothelial cells contribute to tumor angiogenesis via interaction with various growth factors and their cognate receptors (Casali et al., 2022). Therefore, dysregulation of these integrins or mutations of the integrin-related genes can promote the transport of oxygen and nutrition into a tumor through the formation of new blood vessels, which results in the growth of the tumor. Moreover, in response to ECM stiffness, integrin αvβ6 and αvβ8 on cancer-associated fibroblasts (CAFs) are essential to tumor metastasis by activating latent TGF-β1, and promoting the expression of and communication with CAF-related substances, such as platelet-derived growth factor receptor α or β (PDGFRα/β) and pro-inflammatory cytokines (Brown & Marshall, 2019). Additionally, integrins α2β1, α6β1, α6β4, and αvβ3 greatly contribute to the regulation of the stemness-like phenotype of cancer stem cells (CSCs) (Su et al., 2020). Arguably, therapies targeting integrins present tremendous potential for advancing treatments in tumors with higher specificity and better tolerance than traditional medicine. This review focused on the mechanisms and the corresponding targeted therapies of integrin-mediated intercellular crosstalk in tumors.

With rapid advancements in scientific technology and a progressively comprehensive understanding of the contributions of integrins to tumor development, integrin-targeted therapies offer an opportunity to significantly improve patient healthcare. Although suboptimal outcomes in clinical trials have been reported, continued research efforts have been undertaken to enhance current integrin-targeted therapy and innovate anti-tumor treatment strategies.

The intended audience

Researchers studying integrins and tumor biology are major intended audience in this review. This review summarizes the function of integrins in multiple key processes associated with tumor biology via signaling in endothelial cells, cancer-associated fibroblasts, and cancer stem cells. Furthermore, it also includes integrin-mediated targeted therapy and selective drug delivery system for tumor therapeutic strategies. Previous literature less summarizes the integrin-mediated intercellular crosstalk between endothelial cells, cancer-associated fibroblasts, cancer stem cells, and tumor cells. Such interaction plays a pivotal role in tumor growth, angiogenesis, invasion, and metastasis. Additionally, systemic integrin-targeted treatments and precision medicines in tumor biology are attracting increasing attention, but there is little research fully investigating them. Consequently, this review elucidates the impact of integrins on these three distinct cell types in multiple key processes associated with tumor biology and explores the emerging integrin-targeted therapeutic approaches for tumors. In general, the elucidation in integrin-mediated tumor biology and exploitation of novel treatment strategies targeting integrin modulation represent significant breakthroughs in improving patient care, providing new clues for more optimal therapeutic approaches, and promoting comprehensive insights into underlying mechanisms of tumor biology.

Survey Methodology

We conducted an extensive literature review on PubMed, utilizing search terms such as ‘integrin’, ‘tumors’, ‘endothelial cells’, ‘CAFs’, ‘CSCs’, ‘targeted therapy’, and ‘integrin-mediated selective drug delivery system’. This review encompasses research, meta-analysis, and review articles in the English language, but not letters to the editor and case reports. In total, 160 publications were cited in this review published from February 1999 to October 2023.

The effects of integrins on multiple key processes associated with tumor biology

Endothelial cells

Previous studies have revealed that several integrins on endothelial cells regulate tumor angiogenesis, progression, and metastasis, and for lymph angiogenesis (Fig. 2) (Casali et al., 2022; Hakanpaa et al., 2015; Sokeland & Schumacher, 2019).

Figure 2 The pro- and anti-tumorigenic effects of integrins on interaction between endothelial cells and particular tumor cells.

Various integrins communicate with pro- or anti-tumorigenic substances in the tumor microenvironment and other receptors on endothelial cells or specific tumor cells, thus transmitting pro- or anti-tumorigenic signals through regulating FAK, Akt, Erk, MAPK, and NF-κB signaling pathway. Eventually, signals promote or impede matrix stiffness, angiogenesis, lymphangiogenesis, and metastasis of tumors. EC, endothelial cell; LEC, lymphatic endothelial cell; VN, vitronectin; FG, fibrinogen; VEGF, vascular endothelial growth factor; VEGFR2, vascular endothelial growth factor receptor 2; Src, Src family kinase; PI3K, phosphatidylinositol 3-phosphokinase; Akt, protein kinase B; Rac1, Rac family small GTPase; Cdc42, cell division cycle 42; Ca2+, calcium ion; MMRN2, multimerin-2; VCAM-1, vascular cell adhesion molecule 1; CRT, calreticulin; EMILIN-1, recombinant elastin microfibril interface located protein 1; Ang-2, angiopoietin 2; NF-κB, nuclear factor kappa B; FAK, focal adhesion kinase; MAPK, mitogen-activated protein kinase; mTOR, mammalian target of rapamycin; NRP1, n europilin-1; PGE2, prostaglandin E2; COX2, cyclooxygenase-2; MCP1, chemotactic protein 1; PIGF, placenta growth factor; bFGF, basic fibroblast growth factor; CXCL1, chemokine ligand 1; Erk, extracellular regulated protein kinases; MEK, MAPK kinase; uPAR, urokinase-type plasminogen activator receptor; ILK, integrin-linked kinase; EGF, epidermal growth factor; L-T4, L-thyroxine; Tetrac, tetraiodothyroacetic acid; THBS1, thrombospondin 1; HIF1, hypoxia-induced factor 1; OPN, osteopontin; TRAF2, TNF receptor-associated factor 2.

Tumor angiogenesis

In the tumor microenvironment, integrins αvβ3 and αvβ5 on endothelial cells can induce angiogenesis (Casali et al., 2022; Hakanpaa et al., 2015; Korhonen et al., 2016; Lee et al., 2014). For example, hypoxia, a characteristic feature of the tumor microenvironment, can upregulate integrin αvβ3, vascular endothelial growth factor (VEGF), and its receptor tyrosine kinase, vascular endothelial growth factor receptor (VEGFR), via the release of hypoxia inducible factor 1 (HIF1), eventually enhancing angiogenesis (Casali et al., 2022). In breast cancer, hypoxia and osteopontin can form a positive control loop. Hypoxia can induce the expression of osteopontin (OPN), and increased OPN can further promote the expression of HIF1α mRNA and increase HIF1α protein stability by binding to integrin αvβ3 on MDA-MB-231 cells and stimulating the hypoxia-mediated PI3K/integrin-linked kinase (ILK)/Akt pathway and ensuing nuclear factor kappa B (NF-κB) signaling pathway. Furthermore, upregulated HIF1α enhances the expression of VEGF, triggering tumor angiogenesis (Raja et al., 2013). Angiogenesis is an essential process that involves the new blood vessel formation from pre-existing vessels under both physiological and pathophysiological conditions (Casali et al., 2022). During this process, integrins αvβ3, αvβ5, and α5β1 on endothelial cells are upregulated (Bae et al., 2022; Casali et al., 2022; Korhonen et al., 2016; Ruegg, Dormond & Mariotti, 2004). Simultaneously, other cofactors including VEGFR, angiopoietin (Ang), and CD93 transmembrane receptors also participate in angiogenesis. Soldi et al. (1999) indicated that the binding of integrin αvβ3 on endothelial cells with vitronectin or fibrinogen promoted VEGF-induced VEGFR2 phosphorylation (Casali et al., 2022). Subsequently, the phosphorylated VEGFR recruited adaptor proteins, such as Shc, Ras, Src kinase, and tyrosine phosphatases SHP-1 and SHP-2, which further activated the PI3K/Akt and MAPK pathway, resulting in migration of endothelial cells and angiogenesis. Conversely, VEGFR2 can also promote integrin β3 tyrosine phosphorylation via the c-Src inside-out signaling pathway, thus enhancing VEGF-dependent VEGFR2 phosphorylation (Casali et al., 2022; Karaman, Leppanen & Alitalo, 2018). Moreover, integrin αvβ3 on endothelial cells can disrupt the ECM and promote the migration of endothelial cells by binding and activating matrix metalloproteinase 2 (MMP-2) on the migration tip of newly formed blood vessels, and also respond to pro-angiogenic factors such as basic fibroblast growth factor (bFGF), thus facilitating tumor angiogenesis (Rocha et al., 2018). Additionally, in RBE and 9,810 cell lines, lysyl oxidase-like 1 (LOXL1) was overexpressed, which could interact with fibulin 5 and bind to integrin αvβ3. Subsequently, activated downstream signaling pathways of integrin αvβ3 including FAK and MAPK augmented tumor angiogenesis (Yuan et al., 2021). Furthermore, tumor angiogenesis is also affected by the thyroid hormone and tetraiodothyroacetic acid. Schmohl et al. (2020) indicated that in integrin αvβ3 positive SW1736 xenografts, high thyroid hormone aggravated angiogenesis in anaplastic thyroid cancer, whereas low thyroid hormone or tetraiodothyroacetic acid-induced conditions alleviated angiogenesis in integrin αvβ3 negative human hepatocellular carcinoma xenografts. Regarding the specific effect of thyroid hormone and tetraiodothyroacetic acid on tumor angiogenesis, Davis et al. (2014) demonstrated that L-thyroxine (L-T4) converted the soluble actin to fibrous actin and modulated laminin attachment to cells by binding to integrin αvβ3 on endothelial cells, thus regulating endothelial cell migration and angiogenesis. Tetraiodothyroacetic acid suppresses the transcription of epidermal growth factor receptor (EGFR) and basic fibroblast growth factor receptor (bFGFR), which inhibits the pro-angiogenic effect of epidermal growth factor and bFGF, interrupts the communication between VEGFR and integrin αvβ3, thus promoting the expression of thrombospondin 1 (THBS1), the effect of which is anti-angiogenic by binding to integrin αvβ3 on endothelial cells. Davis et al. (2014) also demonstrated that L-T4 binding to integrin αvβ3 on endothelial cells can promote the MMP-9 expression and corresponding pro-invasiveness and pro-metastasis function. Apart from integrin αvβ3-mediated VEGFR2 pro-angiogenic role in multiple solid tumors, Robinson and Hodivala-Dilke indicated that integrin αvβ3 also can contact neuropilin-1 (NRP1), an VEGF co-receptor on endothelial cells augmenting VEGFR2-mediated signaling pathway, thus suppressing VEGF/VEGFR2-induced angiogenesis (Robinson & Hodivala-Dilke, 2011). Consequently, integrin αvβ3 can exhibit both pro-angiogenic and anti-angiogenic effects, which is dependent upon the substances it binds and the specific tumor microenvironment. For example, binding to VEGFR2, vitronectin, fibronectin, Del1, ANGPTL3, CYR61, bone sialoprotein, and thrombin stimulates angiogenesis, whereas binding to THBS1 and tumstatin inhibits it (Hodivala-Dilke, 2008). As a consequence, inhibitors of integrin αvβ3 may display pro-tumorigenic or anti-tumorigenic effects, which may be one of underlying reasons for unsuccess of several integrin αvβ3 antagonists in preclinical or clinical trials, whereas, promoters of anti-angiogenic factor binding can impede tumor angiogenesis and development (Hodivala-Dilke, 2008). Vitronectin and fibronectin with RGD motifs specifically bind to integrin αvβ3 on endothelial cells and enhance integrin αvβ3-mediated angiogenesis. In addition, overexpressed Del-1 in pathological angiogenesis binds integrin αvβ3 on endothelial cells and regulates angiogenesis by inhibiting branching angiogenesis and forming a new, thick, disorganized array of vessels with capillary size. Moreover, thrombin with an RGD motif can upregulate the expression of integrin αvβ3 on endothelial cells and interact with it to enhance angiogenesis. Hodivala-Dilke, Reynolds & Reynolds (2003) indicated that bone sialoprotein containing an RGD motif can also bind integrin αvβ3 on endothelial cells and enhance angiogenesis. Additionally, proteolytic cleavage of collagen type IV results in the exposure of a functional and normally hidden cryptic site that is correlated with angiogenic vessels and the gain of integrin αvβ3 binding, thus also contributing to the enhancement of angiogenesis. Both ANGPTL3 and CYR61 can bind integrin αvβ3 on endothelial cells and induce angiogenic responses in rat corneal pocket assays (Hodivala-Dilke, Reynolds & Reynolds, 2003). Regarding the molecular mechanism of THBS1-mediated anti-tumorigenic effect, Jian et al. (2019) demonstrated that, in xenograft tumor model, THBS1 binding to CD36 on microvascular endothelial cells can trigger intrinsic and extrinsic apoptotic pathways to augment endothelial cell apoptosis, and suppress endothelial cell proliferation, migration and tube formation in osteosarcoma. Subsequently, a phase I clinical trial studying the safety of the conjugation of bevacizumab, a VEGF antagonist, and ABT-510, THBS1 agonist, found that, among the thirty-four patients with diverse cancer types, six experienced clinical benefit from this treatment (Lawler, 2022). In addition, Hamano & Kalluri (2005) pointed out that the amino acids 185-203 of tumstatin binding to integrin αvβ3 activates FAK/PI3K and inhibits melanoma cell proliferation, whereas, amino acids 54-132 of tumstatin dephosphorylates FAK/PI3K/Akt/ mammalian target of rapamycin (mTOR) on endothelial cells and induces endothelial cell apoptosis. Moreover, tumstatin also impedes the expression of cap-dependent protein in the proliferating endothelial cells, which is due to the augmented coupling of eukaryotic initiation factor 4E protein (eIF4E) with 4E-binding protein 1. In in vivo and in vitro studies conducted by Thevenard et al. (2010) revealed that the YSNSG cyclopeptide, derived from tumstatin, alters endothelial cell migration, can modulate the distribution of β1-integrin within endothelial cell lamellipodia, dephosphorylate FAK, and significantly diminish the number of lamellipodia, ultimately, alleviating tumor angiogenesis and migration. With the above mentioned, we can find that severe hypoxia may upregulate integrin αvβ3 and enhance its pro-angiogenic activity through HIF1α, while mild hypoxia may induce THBS1 secretion and activate the anti-angiogenic pathway. Meanwhile, high stiffness ECM enhances the synergistic effect of integrin αvβ3 with VEGFR2 through integrin mechano-signaling, whereas THBS1 binds integrin αvβ3 more readily in low stiffness environments. In addition, specific tumor models also influence the effect of integrin αvβ3 on tumor vasculature. For example, in MDA-MB-231 cells, hypoxia promotes angiogenesis by enhancing integrin αvβ3-VEGFR2 interactions through upregulation of HIF1α, whereas THBS1 overexpressed breast cancer models show integrin αvβ3-mediated vasopressor. In integrin αvβ3-negative hepatocellular carcinoma, thyroid hormones inhibit angiogenesis by modulating MMP-9 and EGFR. Consequently, the dichotomous roles of integrin αvβ3 in tumor angiogenesis are highly context-dependent, influenced by tumor model specificity and dynamic tumor microenvironment regulation.

Integrin α5β1 is associated with VEGF and Ang2, which are involved in matrix stiffness-induced angiogenesis. Dong et al. demonstrated that matrix-stiffness signals were transduced by the stiffness sensor integrin α5β1 in hepatocellular carcinoma (HCC) cells, further activating the PI3K/Akt signaling pathway, increasing expression of VEGF, and eventually causing lymph angiogenesis and HCC metastasis (Dong et al., 2014). Previous studies have shown that stiffness mechanical signals transmitted by integrin α5β1 upregulated VEGFR2 in human umbilical vein endothelial cells and VEGF expression in HCC cells to promote stiffness-induced angiogenesis (Li et al., 2022a). Notably, Ang is over-expressed on tumor cells and endothelial cells, and Ang2 can act as a weak agonist of its cognate receptor Tie2 on endothelial cells to induce the sprouting of endothelial cells and increased endothelial cell permeability (Hakanpaa et al., 2015; Korhonen et al., 2016; Lee et al., 2014). Lee et al. (2014) proposed that integrin α5β1 on endothelial cells can bind to Ang2 via Gln-362 in Ang2 and tail sections of α5 in integrin α5β1, thus activating the downstream FAK, ILK, Akt, and extracellular regulated protein kinases (Erk) signaling pathways. Therefore, the connection between integrin α5β1 and Ang2 enhances the destabilization of endothelial cells and promotes the differentiation and migration of Tie2-negative tip cells for sprouting, thus facilitating tumor cell metastasis in breast cancer (Imanishi et al., 2007; Lee et al., 2014). Furthermore, Hakanpaa et al. (2015) demonstrated that Ang2-integrin α5β1 interactions in Tie2-negative cells activated PI3K/Akt and Erk signals and inhibited the phosphorylation of Rho signals to maintain the actin skeleton and formation of actin stress fibers and promote trans endothelial tumor cell migration in vitro. Apart from the direct effects, integrin α5β1 and α3β1 work with CD93 transmembrane receptors, calcium-mediated macropinocytosis and lysosomal exocytosis in endothelial cells to promote tumor angiogenesis (Bae et al., 2022; Langenkamp et al., 2015; Lugano et al., 2018). Lugano et al. (2018) showed that CD93 was specifically expressed and localized in filopodia of differentiated tip cells anchored by multimerin-2 (MMRN2). Furthermore, CD93 is also crucial for integrin α5β1 activation and fibronectin fibrillogenesis during high-grade glioma angiogenesis. Therefore, crosstalk of α5β1 and CD93 improves tube formation, migration of tip cells and endothelial cells, and actin skeleton organization (Langenkamp et al., 2015; Lugano et al., 2018). Similar to integrin αvβ3, integrin α5β1 has both pro-tumorigenic and anti-tumorigenic effect as well. Several studies demonstrated that endostatin extracted from hemangioendothelioma binds to nucleolin and integrin α5β1 on endothelial cells, further being translocated into the nucleus of endothelial cell by the integrin α5β1-nucleolin-urokinase-type plasminogen activator receptor (uPAR) complex. Such internalization of α5β1, nucleolin, and uPAR in endothelial cells inhibits focal adhesion and its downstream signaling pathway, such as MEK/Erk and PI3K/Akt, which further impeding NF-κB translocation and chemokine (C-X-C motif) ligand 1 (CXCL1) expression. Suppression in CXCL1 transcription can decrease the number of monocyte chemotactic protein 1 (MCP1), placenta growth factor (PIGF), and bFGF, thus alleviating its pro-angiogenic effect in in vitro and in vivo studies of hemangioendothelioma and hemangiosarcoma, and hemangiopericytoma (Guo et al., 2015; Song et al., 2012b). Bae et al. (2022) has shown that integrin α3β1 expression on glioblastoma endothelial cells boosted the influx of extracellular calcium via contacting with calreticulin, while simultaneously activating the Rac1/Cdc42 signaling pathway, promoting micropinocytosis, exocytosis of macropinosomes, and driving capillary tube formation during glioblastoma angiogenesis (Bae et al., 2022). Moreover, Yang et al. (2021) found that, T7 peptide, a part of tumstatin, can interact with integrin α3β1 and decrease the expression of cyclooxygenase-2 (COX2) and prostaglandin E2 (PGE2) under normoxic conditions, which inhibits endothelial cell proliferation, migration and tube formation, and augments endothelial cell apoptosis, ultimately exerting anti-angiogenic effect in HCC mouse models.

In conclusion, integrins αvβ3, αvβ5, α5β1 on endothelial cells critically regulate tumor angiogenesis, with hypoxia amplifying αvβ3 and VEGF/VEGFR pathways. While αvβ3 exhibits dual pro-/anti-angiogenic roles via ligand-specific signaling, α5β1 drives matrix stiffness-mediated angiogenesis with Ang2/CD93. Clinical challenges arise from context-dependent integrin signaling, causing inconsistent αvβ3 inhibitor outcomes. Future strategies prioritize isoform-specific inhibitors, combination therapies against resistance, biomarker-guided precision medicine, and deeper exploration of tumor microenvironment integrin dynamics.

Lymph angiogenesis

Integrins α4β1, α6, and α9β1 on lymphatic endothelial cells (LECs) can trigger lymph angiogenesis, thereby inducing tumor metastasis in the tumor microenvironment (Karaman, Leppanen & Alitalo, 2018). Lymph angiogenesis consists of proliferation, migration, invasion of LECs, and tube formation to form lymphatic capillaries and collecting lymphatic vessels (Capuano et al., 2019; Danussi et al., 2013; Karaman, Leppanen & Alitalo, 2018; Nishino et al., 2021; Ren et al., 2022). It has been demonstrated that vascular endothelial growth factor A (VEGFA), vascular endothelial growth factor C (VEGF-C), vascular endothelial growth factor-D (VEGFD), and vascular endothelial growth factor receptor 3 (VEGFR3) are involved in lymph angiogenesis (Karaman, Leppanen & Alitalo, 2018). Nishino et al. (2021) first demonstrated that VEGFD was highly released from lymphangioleiomyomatosis (LAM) cells, and bound to VEGFR3 and integrin α9 in the LEC plasma membrane, thereby inducing the PI3K/Akt pathway and lymph angiogenesis. Both VEGFD/VEGFR3 and VEGFD/integrin α9 trigger LEC proliferation and migration. Furthermore, EMILIN-1 in the tumor microenvironment also plays a critical role in lymph angiogenesis, similar to that of VEGFC and VEGFD (Capuano et al., 2019). Capuano et al. (2019) revealed that the gC1q domain of EMILIN-1 bound to integrin α4β1 on LECs for proliferation and tube-like structure formation, and to integrin α9β1 on LECs to form collecting lymphatic vessels (Danussi et al., 2013). In addition, it has been shown that integrin α6 in lung adenocarcinoma (LUAD) tissues overexpression facilitates K63 polyubiquitination of TNF receptor-associated factor 2 (TRAF2) to maintain the activity of nuclear factor-κB (NF-κB) signaling pathway in a popliteal lymph node metastasis model, ultimately resulting in increased microlymphatic vessel density and lymphatic metastasis in LUAD (Ren et al., 2022).

In summary, lymphatic integrins (α4β1, α6, α9β1) promote lymph angiogenesis/tumor metastasis via VEGFD/EMILIN-1 interactions, whereas specific signaling pathways remain unclear. Future studies using single-cell analysis should identify therapeutic targets for anti-metastatic clinical trials.

Metastasis

Integrin αvβ3, α3β1, α4β1, α5β1, and α6β1on endothelial cells or tumor cells can induce tumor migration and metastasis in the tumor microenvironment (Sokeland & Schumacher, 2019). Many types of tumors have a poor prognosis due to migration and metastasis. Metastasis comprises dissociation from the primary tumor, disrupting the basement membrane of tumor vessels, intravasation, survival in circulation, adhesion to endothelial cells and extravasation (Chen et al., 2016a; Sokeland & Schumacher, 2019). Studies focusing on metastasis have shown that, through the epithelial-mesenchymal transition (EMT) process, neoplastic cells gain migratory capacity that facilitates their detachment from the primary tumor mass (Sokeland & Schumacher, 2019). Regarding the extravasation phase of metastasis, it comprises rolling, adhesion, trans endothelial migration (TEM)/diapedesis, and basement membrane breaching (Chen et al., 2016a; Sokeland & Schumacher, 2019). After selectin-mediate rolling, integrins on tumor cells, leukocytes, or endothelial cells are activated and prepared for adhesion (Benedicto et al., 2017; Sokeland & Schumacher, 2019). Sokeland and Schumacher emphasized that, in prostate cancer and melanoma, integrins αvβ3 and α5β1 expressed on tumor cells bound to L1-CAM on endothelial cells (Gavert et al., 2008; Sokeland & Schumacher, 2019). In contrast, colorectal carcinoma, renal clear cell carcinoma, pancreatic ductal adenocarcinoma (PDAC), and breast cancer express L1-CAM to bind to integrin α5β1 (VLA-5) comprehensively and to integrin αvβ3 partially on endothelial cells (Allory et al., 2005; Gavert et al., 2005; Sebens Müerköster et al., 2007; Sokeland & Schumacher, 2019). In addition, integrin α4β1 (VLA-4) on lymphoma, myeloma, and oral squamous cell carcinoma adhere to VCAM-1 on endothelial cells to promote metastasis (Sanz-Rodríguez & Teixidó, 2001; Schlesinger & Bendas, 2015; Song et al., 2012a). Furthermore, VLA-4 participates in tumor lymph angiogenesis, thereby enhancing lymph invasion (Garmy-Susini et al., 2010). However, multiple tumors, such as melanoma, oral squamous cell carcinoma, and colorectal cancer, express intracellular adhesion molecule-1 (ICAM-1) rather than integrin, and endothelial cells also express ICAM-1; consequently, leukocytes with integrin αLβ2 (LFA-1) act as a bridge between tumor and endothelial cells (Liang et al., 2007; Sokeland & Schumacher, 2019; Usami et al., 2013). To produce TEM/diapedesis, Ishikawa et al. (2014) reported that, in solid-phase binding assays, integrins α3β1, α6β1, αvβ3,and α6β4 on glioma cells can interact with laminin 411, 421, and 332 in the subendothelial ECM, respectively (Ren et al., 2022). Mechanistically, the activated integrin β1 activates an intracellular signal, regulates actin stress fibers, and forms invadopodia, eventually sustaining TEM, promoting a breach of the basement membrane breaching and resulting in melanoma, glioma, fibrosarcoma, and prostate, lung, colon, pancreas, breast and tongue cancer migration (Ishikawa et al., 2014; Ren et al., 2022). In addition to extravasation, integrin β1 also contributes to metastatic colony formation (Ren et al., 2022). Research has shown that integrin β1 mediates CXCL1-induced-FAK/Akt pathway activation and promotes MMP-2/9 expression, thereby enhancing gastric cancer metastasis to lymph nodes (Wang et al., 2017). Additionally, Chen et al. (2008) conducted CHO transfectants overexpressing ADAM15, which is preferentially produced in multiple aggressive tumors. They reported that ADAM15 can inactivate Erk1/2 and cause expression and clustering of integrin α5 rather than β1 on CHO cells, eventually repressing B16F10 melanoma cells lung metastases (Chen et al., 2008).

To sum up, integrins αvβ3/α5β1 drive tumor metastasis via mechanisms like extravasation, but targeting these receptors remains difficult. Research should prioritize β1’s involvement in intracellular signaling, actin dynamics, and invadopodia. Blocking CXCL1-triggered FAK/Akt pathways may suppress metastatic spread.

Cancer-associated fibroblasts

Beyond endothelial cells, integrins also modulate CAFs effect on multiple key processes associated with tumor biology. It has been reported that integrin-dependent activation, differentiation, secretion and metabolism of CAFs contribute significantly to tumor progression, metastasis, and invasion (Fig. 3) (Brown & Marshall, 2019; Deng et al., 2022; Hupfer et al., 2021; Jang & Beningo, 2019; Peng et al., 2018; Wang et al., 2014; Zhang et al., 2020).

Figure 3 Communication between cancer-associated fibroblasts and specific tumor cells via integrins.

Various integrins contact proteins in the tumor microenvironment and other receptors on cancer-associated fibroblasts, thus transmitting signals through FAK, JNK, MAPK, and NF-κB signaling pathways. Consequently, signals promote cancer-associated fibroblast differentiation, tumor progression, and invasion. CAF, cancer-associated fibroblast; SDF-1, stromal cell-derived factor-1; CXCR4, C-X-C chemokine receptor type 4; FN, fibronectin; OPN, osteopontin; LN-332, laminin 332; IL-6, interleukin 6; IL-32, interleukin 32; THBS2, thrombospondin 2; NF-κB, nuclear factor kappa B; LOXL1, lysyl oxidase-like 1; PDGF-BB, platelet-derived growth factor; PDGFRβ, platelet-derived growth factor receptor β; TGF-β, transforming growth factor β; MMP, matrix metalloproteinases; FAK, focal adhesion kinase; JNK, c-Jun N-terminal kinase; MAPK, mitogen-activated protein kinase; PDAC, pancreatic ductal adenocarcinoma; CRC, colorectal cancer; GC, gastric cancer; EMT, epithelial-mesenchymal transition; α-SMA, α -smooth muscle actin; AMPK, AMP-activated kinase; p-Smad2/3, phosphorylated drosophila mothers against decapentaplegic protein 2/3; Gal-1, galectin-1; Gli, glioma-associated oncogene 1; PLC, phospholipase C; PKC, protein kinase C; Src, Src family kinase; Erk, extracellular regulated protein kinases.

CAF activation

Integrins αvβ6 and αvβ8 play a significant role in CAF activation by activating transforming growth factor β (TGF-β). CAFs originate from dormant resident fibroblasts in the tumor stroma, pericytes, and bone marrow-derived mesenchymal stem cells (Nan et al., 2022; Zeltz et al., 2020). Activation of CAFs plays a pivotal role in tumor progression, metastasis, and invasion through integrin αvβ6/αvβ8-mediated TGF-β activation (Brown & Marshall, 2019; Peng et al., 2018). Peng et al. (2018) demonstrated that integrin αvβ6 was highly expressed on colorectal cancer (CRC) cells and activated latent TGF-β in the ECM and on the cell surface (Zeltz et al., 2020). Brown & Marshall (2019) also found that integrin αvβ8 on H1264 lung cancer cell lines activated latent TGF-β in the tumor microenvironment via MMP-14 (Zeltz et al., 2020). Jang & Beningo (2019) revealed that laminin-332 secreted from CAFs bound to integrin α3β1 on CAFs and played an essential role in TGF-β-induced CAF differentiation and PDAC invasion. After CAF activation, the upregulation of α-smooth muscle actin (α-SMA), fibroblast activation protein (FAP), PDGFRα/β, secretory proteins, and cytokines may stimulate the EMT, tumor progression, and metastasis (Brown & Marshall, 2019; Nan et al., 2022; Peng et al., 2018). Peng et al. (2018) also showed that activated CAFs can secrete stromal cell-derived factor-1 (SDF-1) and stimulate the SDF-1/ C-X-C chemokine receptor type 4 (CXCR4)/integrin αvβ6 axis to promote CRC progression. In addition, CAFs secrete MMP-3 and MMP-9 to degrade the ECM and promote tumor invasion and metastasis (Deng et al., 2022). Nan et al. (2022) reported that TGF-β-activated CAFs can activate the phosphorylated drosophila mothers against decapentaplegic protein 2/3 (p-Smad 2/3) pathway and increase expression of thrombospondin 2 (THBS2), which then bind to integrin αvβ3 in PDAC cells and activate the MAPK pathway, eventually enhancing PDAC growth. Furthermore, CAFs can induce desmoplasia and upregulate integrins α5β1 and αvβ5 on CAFs, finally promoting PDAC growth (Deng et al., 2022; Zeltz et al., 2020). Ji et al. (2020) indicated that, in a CRC model, CRC extracellular vehicles (EVs) rich in integrin β-like 1 activated tumor necrosis factor α-induced protein 3 (TNFAIP3) mediated the NF-κB signaling pathway to activate CAFs, and then the CAFs released pro-inflammatory cytokines such as interleukin-6 (IL-6) and interleukin-8 (IL-8) to promote CRC metastasis. Consequently, Jianpi Jiedu Recipe (JPJDR), a medication targeting EVs rich in integrin β-like 1, adopted the same mechanism to decrease the activation of CAF, thereby inhibiting CRC metastasis (Li et al., 2022b).

In a nutshell, activated CAFs, driven by integrins αvβ6/β8 and TGF-β, enhance tumor progression through ECM remodeling and cytokine secretion. Key challenges include CAF heterogeneity, off-target effects, and therapy resistance. Emerging strategies focus on subtype-specific markers, integrin inhibitors, combination therapies, and precision drug delivery. Promising approaches involve extracellular vesicles, metabolic reprogramming targeting, predictive biomarkers, and personalized treatment models to optimize CAF-directed anticancer interventions.

The interaction of integrin and upregulated proteins induced by CAF activation

The interplay of integrin α11β1 on CAFs and the overexpressed proteins after CAF activation, such as α-SMA, PDGFRα and β, and variable cytokines on the cell surface and ECM, can induce desmoplasia and the progression, migration, and invasion of breast cancer (BC), non-small cell lung cancer (NSCLC), PDAC, and head and neck squamous cell carcinoma (HNSCC) (Deng et al., 2022; Hupfer et al., 2021; Jang & Beningo, 2019; Primac et al., 2019; Zeltz et al., 2022; Zeltz et al., 2020). Previous studies have shown that the interaction of fibrillar collagen receptor integrin α11β1 and PDGFRβ contributed to matrix stiffness and collagen deposition (Deng et al., 2022; Primac et al., 2019; Zeltz et al., 2020). Furthermore, the rigid ECM provides powerful physical support for tumor growth and adhesion (Deng et al., 2022). Highly expressed integrin α11β1 on CAFs activates PDGFRβ and its downstream Jun N-terminal kinase (JNK) signaling in response to PDGF-BB to support BC, NSCLC, and PDAC progression and invasion. Moreover, integrin α11β1 also increases secretion of insulin-like growth factor 2 (IGF-2) and lysyl oxidase (LOX) family members (Jang & Beningo, 2019; Primac et al., 2019; Zeltz et al., 2022; Zeltz et al., 2020). For example, α11β1 in NSCLC-derived CAFs induced the lysyl oxidase-like 1 (LOXL1) secretion, an ECM cross-linking enzyme, to initiate collagen deposition, matrix stiffness, and tumor invasion (Deng et al., 2022; Zeltz et al., 2022; Zeltz et al., 2020). Zeltz et al. (2020) stated that LOXL2 expressed from BC played a pivotal role in CAF activation and α-SMA expression via phosphorylating integrin β1-mediated FAK signaling pathway. Moreover, in a BC mouse model, inhibiting FAK cause impaired tumor growth and infiltration of leukocytes and macrophages. Jang & Beningo (2019) demonstrated that the correlation of integrin α11β1 on CAFs and α-SMA promoted fibrillar collagen assembly, ECM remodeling, and tumor metastasis in NSCLCs, which can be presented by exacerbated tumorigenicity of NSCLC in a co-cultured model with mouse embryonic fibroblasts (MEFs) expressing integrin α11. Zeltz et al. (2022) also indicated that integrins can be activated in the presence of non-integrin adhesion receptors such as syndecans in CAFs. In BC, syndecan-1 on CAFs coordinates with and activates integrin αvβ3 to enhance fibronectin (FN) reorganization. Similarly, syndecan-4 on CAFs detects mechanotension and activates integrin β1, thereby promoting FN assembly and stimulating RhoA, ultimately contracting acto-myosin and aggravating matrix stiffness (Zeltz et al., 2022). Therefore, integrin α11β1 controls tumor desmoplasia in PDGFRβ+, α-SMA+, and syndecan+ subsets of CAFs. Matrix stiffness also induces matrix autophagy via integrin αv/FAK/AMP-activated kinase (AMPK) signaling, eventually triggering the formation of a pro-tumorigenic niche (Hupfer et al., 2021).

FN not only contributes to matrix stiffness, but also engages in directional migration, invasion, and metastasis of prostate, pancreatic, and colon tumors along FN matrices and protrusion of CAFs (Attieh et al., 2017; Erdogan et al., 2017; Jang & Beningo, 2019; Miyazaki et al., 2020; Zeltz et al., 2020). Integrin αvβ1 on CAFs participates in the fibrillar fibronectin assembly of CAFs via PDGFRα, and in myosin light chain 2 (MLC2) contractility and traction forces, which then induces directional mobility of tumor cells (Attieh et al., 2017; Erdogan et al., 2017; Jang & Beningo, 2019; Zeltz et al., 2020). However, the presence of integrin αvβ1 is determined by integrin αvβ5-regulated αvβ1 endocytosis by CAFs and desmoplasia in PDAC (Zeltz et al., 2020). Additionally, in colon and pancreatic tumors, integrin αvβ3 assists αvβ1 to produce parallel fibronectin alignment that mediates communication between αvβ1 on tumor cells and assembled fibronectin on the CAF surface, thus triggering tumor migration in a specific direction (Jang & Beningo, 2019; Zeltz et al., 2020).

Furthermore, activated CAFs release SDF-1/CXCL12, OPN, periostin, galectin, THBS2, MFGE8, and diverse cytokines such as IL-6 and IL-32, which promote the EMT, tumor invasion, and tumor metastasis (Jang & Beningo, 2019; Qin et al., 2018; Wen et al., 2019). CAF-secreted IL-32 with an RGD motif binds to integrin β3 on BC cell surface and activates downstream p38 MAPK signaling pathway, thus increasing the expression of several EMT markers including FN, N-cadherin, and vimentin (Jang & Beningo, 2019; Wen et al., 2019). Similarly, IL-6 from CAFs also promotes the EMT and esophageal adenocarcinoma invasion (Jang & Beningo, 2019). Moreover, IL-6 stimulates the secretion of OPN from CAFs in head and neck cancer (HNC) to enhance HNC progression via the NF-κB pathway (Qin et al., 2018). OPN secreted from CAFs communicates with integrin α9β1 in BC, promoting the recruitment of CAFs and BC lymphatic metastasis in a xenograft mouse model in vivo (Ota et al., 2014; Qin et al., 2018). In the SDF-1/CXCR4 axis, integrin αvβ6 is overexpressed in tumor cells due to Erk phosphorylation and stimulation of the Ets-1 transcription factor, ultimately boosting CRC, BC, and prostate cancer metastasis to the liver, lungs, and lymph nodes in a co-culture of the human normal colonic fibroblast cell line CCD-18Co and human CRC cell line HT-29 or RKO assay (Peng et al., 2018; Wang et al., 2014). The SDF-1/CXCR4 axis also improves the clustering of integrin β1 in GC cells and promotes FAK signaling, thereby accelerating gastric cancer (GC) invasion (Izumi et al., 2016). Several studies have shown that in GC, the cooperation of integrin β1 in MGC-803 cells and galectin-1 (Gal-1) derived from CAFs increases the expression level of glioma-associated oncogene 1 (Gli1), activates hedgehog (Hh) signaling pathway, and facilitates the EMT and GC invasion (Chong et al., 2016; He et al., 2014). Moreover, research has found that the expression of galectin-3 is proportional to that of integrin αvβ1 colocalized with CD63 in metastatic tumor cells, and accelerates FN reassembly and BC metastasis in a mouse model of BC (Zhang et al., 2022a). Regarding THBS2, several studies demonstrated that it can increase the expression of MMP13 and MMP9, further activating integrin αvβ3/FAK/Akt/NF-κB and integrin αvβ3/phospholipase C (PLC)/protein kinase C (PKC)/c-Src/NF-κB signaling pathway, respectively, which ultimately facilitating lung cancer and osteosarcoma metastasis (Liu et al., 2020; Liu et al., 2018). With respect to MFGE8, Liu et al. (2023a) found that it can aggravate angiogenesis, metastasis, and progression in esophageal squamous cell carcinoma (ESCC) via binding to integrin αvβ3 and integrin αvβ5 in HUVECs and phosphorylating PI3K/Akt/STAT3 and Erk/Akt signaling pathway in in vitro study.

To summarize, integrin α11β1 on CAFs drives tumor progression via α-SMA/PDGFR interactions, inducing desmoplasia and ECM remodeling. It upregulates IGF-2/LOXL1 secretion, enhancing matrix stiffness and invasion. Despite CAF heterogeneity and therapy resistance challenges, targeting specific subtypes with selective inhibitors, combination therapies, and tumor microenvironment modulation could improve outcomes by optimizing drug delivery.

CAF metabolism

Aside from CAF activation and secretion, CAF metabolism, known as the reverse Warburg effect, has been intensely studied in recent years. Sung et al. (2020) showed that upregulated integrin α4 in triple-negative breast cancer (TNBC) cells promoted aerobic glycolysis and BCL2 interacting protein 3 like (BNIP3L)-dependent mitophagy in CAFs to provide significant energy via the reverse Warburg effect, thus furthering BC progression in co-culture assays. At the mitochondrial level, Zhang et al. (2020) revealed that integrin β2 on CAFs can facilitate NADH oxidation in the mitochondrial oxidative phosphorylation system via the PI3K/Akt/mTOR axis, thereby increasing oral squamous cell carcinoma proliferation in vitro and in vivo study.

In brief, recent studies regarding CAFs highlight their metabolic tumor fueling via the reverse Warburg effect. Key challenges: unclear mechanisms and therapeutic targeting. Future aims: clarifying molecular pathways and developing precision therapies.

Cancer stem cells

Numerous studies have shown that multiple types of integrins regulate the properties and behaviors of CSCs, including self-renewal, invasiveness, adhesion, proliferation, and apoptosis (Fig. 4). These findings highlight the importance of integrins in CSC biology and their potential as therapeutic targets. Below, we summarize recent research on the role of integrins in various cancer types, focusing on their mechanisms of action and implications for tumor progression and therapy resistance.

Figure 4 Summary diagram of the interplay between integrins and cancer stem cells in specific tumor types.

Integrins modulate the progression of specific tumors positively through multiple classical signaling pathways, including PI3K, FAK/Akt, ErbB2, c-Met, and PI3K/Akt/NF-κB. FAK, focal adhesion kinase; Erk, extracellular regulated protein kinase; Akt, protein kinase B; PI3K, phosphatidylinositol 3-phosphokinase; ErbB2, epidermal growth factor receptor family; c-Met, cellular-mesenchymal epithelial transition factor; TAZ, Tafazzin; NF-κB, nuclear factor kappa B; p38 MAPK, mitogen-activated protein kinase subfamily; Collagen I, type 1 collagen; VEGFR2, vascular endothelial growth factor receptor 2; LN, laminin; FSTL1, human follistatin-like protein 1; ZEB1, zinc finger E-box binding homeobox 1; TAp63α, tumor protein 63 isoform 1; CSC, cancer stem cell; TAM, tumor associated macrophage; TET3, ten-eleven translocation enzyme 3; 5mC, DNA 5′methylation; GSC, glioma stem cell; NK cell, natural killer cell; WISP1, Wnt-induced signaling protein 1; MMP, matrix metalloproteinase; HIFα, hypoxia inducible factor α; TGF-β, tumor growth factor β; Cyr61, cysteine-rich 61.

Breast cancer and integrins

Multiple studies have reported that integrins may participate in the regulation of breast cancer stem cells. Recently, Barnawi et al. (2019) found that the upregulation of integrin β1 mediated by fascin in breast cancer cells was completely dependent on and contributed to the enrichment of breast CSCs. Moreover, they detected that the fascin/integrin β1 axis promoted the self-renewal of breast CSCs partially through FAK. Specific cell surface markers, including CD44 and CD24 have been widely used to identify certain mesenchymal carcinoma cell populations that are often more intractable and invasive due to enrichment of CSCs (Nieto et al., 2016). However, Bierie et al. (2017) found that CD44 and CD24 alone could not identify CSC-enriched mesenchymal subpopulations because integrin β4 expression levels were similar across these groups when using CD44/CD24 markers. Moreover, they also identified that, for TNBC, the mesenchymal subpopulation with high integrin β4 expression showed increased presentation of CSCs and may contribute to cancer relapse post-treatment. They also detected a zinc finger E-box binding homeobox 1 (ZEB1)-tumor protein 63 isoform 1 (TAp63α)-integrin β4 axis that regulated the expression of integrin β4, thus influencing the pathophysiological behavior of CSCs in a high mesenchymal state. In TNBC, integrin α6β1 associated with the VEGF receptor 2-neuropilin 2 complex, which promoted the activation of FAK and Erk and subsequently the transcription of Hedgehog pathway components (Goel et al., 2012). In mesenchymal-like breast CSCs, the communication between the ECM and integrin α6β1 also promotes the activation of TAZ, inducing a self-renewal and tumor-initiation program that includes overexpression of its ligand laminin 511 (Chang et al., 2015). In contrast to the pathway of integrin α6β1, integrin αv also has the tumor-initiation potential achieved by regulating the expression of Slug independently of FAK in basal breast CSCs (Desgrosellier et al., 2014). Apart from these findings, integrin α6 has been identified to be associated with a poor prognosis in breast cancers based on the presence of a breast cancer stem cell (BCSC) subpopulation. Researchers validated its practicability as a biomarker in predicting the recurrence of breast cancer based on the prevalence of BCSCs (Qiu et al., 2019). Moreover, it was also reported that integrin α9 promoted tumor growth and metastasis in TNBC. Wang et al. (2019) demonstrated that an integrin α9 knockout (KO) suppressed the CSC-like property of TNBC cells and influenced other tumorigenic processes. They confirmed that the integrin α9 KO relocated ILK from the membrane to the cytoplasm, where ILK interacted with PKA to suppress its activity, ultimately leading to elevated glycogen synthase kinase 3 (GSK3) activity, promoting the degradation of β-catenin and influencing the CSC-like property of TNBC cells. Moreover, research has shown that the FSTL 1/integrin β3/Wnt/β-catenin signaling axis regulates the development and chemoresistance of BCSCs (Cheng et al., 2019).

Glioma and integrins

Recently, the correlation between integrins and the regulation of glioma stem cells (GSCs) has also been widely reported. The transcription factor KLF4 directly binds to the promoter of integrin β4, facilitating its transcription and leading to increased expression of integrin β4 in glioma, thus forming a positive feedback loop that promotes glioblastoma stem cell self-renewal and gliomagenesis (Ma et al., 2019). Notably, the increased expression of integrin β4 allows it to bind KLF4 while simultaneously weakening its interaction with its E3 ligase, the von Hippel-Lindau protein, subsequently decreasing KLF4 ubiquitination and leading to its accumulation (Ma et al., 2019). Generally, GSCs secrete the Wnt-induced signaling protein 1 (WISP1) to facilitate a pro-tumor microenvironment by promoting the survival of both GSCs and tumor-associated macrophages. WISP1 signals through the integrin α6β1-Akt pathway to maintain GSCs and M2 TAMs separately in an autocrine and paracrine manner (Tao et al., 2020). Moreover, the binding of laminin and integrin α6β1 can facilitate adhesion to the abluminal surface of the endothelial basement membrane and transmit self-renewal signals through FAK (Lathia et al., 2010). In more immature subpopulations of glioblastoma and esophageal carcinoma stem cells, the activation of FAK and invasive outgrowth depend on the binding of laminin with integrin α7β1 (Haas et al., 2017). Recently, Herrmann et al. (2020) demonstrated that activating the integrin α6-FAK signaling pathway induced the activation of STAT3. Activated STAT3 combines with the promoter of ten-eleven translocation enzyme 3 (TET3) dioxygenase, and then the STAT3/TET3 complex binds to DNA 5′methylation (5 mC), which in turn upregulates certain genes significant for the GSC phenotype. Silencing STAT3, TET3, or both reduces the accumulation of 5 hmC and thus represses the expression of certain genes critical for the maintenance, survival, proliferation, and therapy-resistance of GSCs, such as c-Myc, BclXL, and Survivin (Herrmann et al., 2020). Integrins can also mediate cancer progression by interfering with the interactions between GSCs and other cells. When the blood-brain barrier is disrupted by a tumor, NK cells enter the glioma tumor tissue and interact with glioblastoma stem cells, inducing both the release and production of TGF-β by GSCs in an intercellular crosstalk-dependent manner, in which the interaction between integrin αv on GSCs and CD9 and CD103 on NK cells is necessary. Then, TGF-β is cleaved to become its biologically active form by proteases, such as MMP-2 and MMP-9, which are released mainly by GSCs. Moreover, the release of these MMPs is further driven by integrin αv and by TGF-β itself. Next, TGF-β irreversibly inhibits the cytotoxic function of NK cells by inducing changes in their phenotype, transcription factors, cytotoxic molecules, and chemokines, thus helping GSCs evade NK cells and contributing to the progression of glioma (Shaim et al., 2021).

Prostate cancer and integrins

In prostate cancer, integrin α2β1 has been shown to inhibits cell proliferation while promoting migration and invasion by enhancing the phosphorylation of p38 MAPK (Ojalill et al., 2018). Generally, integrin β4 solely pairs with integrin α6 acting as a receptor for the basement membrane protein laminin. It has been widely reported that integrin β4 is involved in the PI3K, FAK/Akt signaling pathway to regulate tumor progression. Research has shown that targeted deletion of the signaling domain of integrin β4 impaired the self-renewal capacity of prostate tumor progenitors and the expansion of their transit-amplifying derivatives by interrupting ErbB2 and c-Met signaling pathways (Yoshioka et al., 2013). Moreover, it was reported that integrin α2 and EZH2 had low expression in prostate cancer but can be considered as a marker of prostate CSCs due to their distinguishable expression levels (Hoogland et al., 2014).

Colorectal cancer and integrins

In addition to prostate cancer, CRC is also a popular topic. Combined with type I collagen, integrin α2β1 activates PI3K/Akt signaling, thus enhancing the stemness and metastasis of CSCs (Wu et al., 2019). This pathway is highly associated with the drug resistance of colorectal CSCs (Dai, Hu & Zheng, 2017). In colorectal cancer, integrin β1 mediates the dedifferentiation of CD133-negative colorectal cancer cells to generate CSCs. During this process, the ECM regulates cytoskeletal F-actin bundling through biomechanical force associated integrin β1, leading to the degradation of the glycolytic rate-limiting enzyme phosphofructokinase by releasing the E3 ligase tripartite motif protein 11. Ultimately, HIF1 promotes the reprogramming of transcription factors correlated with stem cells, facilitating cancer cell dedifferentiation to generate CSCs (Han et al., 2022). Furthermore, it was demonstrated that cysteine-rich 61 (Cyr61), preferentially expressed in adipose-derived stem cells (ADSCs), combined with its functional receptor integrin αvβ5 to activate downstream FAK/ NF-κB signaling and FAK/HIF-α/STAT3/MMP-2 signaling, thereby promoting tumor growth and metastasis, especially in CRC progression (Liang et al., 2021).

Other cancers and integrins

Integrins have also been implicated in less-studied cancer types. Higher expression of integrin α7 in tongue squamous cell carcinoma (TSCC) is often accompanied by an advanced state of cancer and higher expression of CSC markers. Knockdown of integrin α7 inhibited the proliferation and stemness of cancer cells but promoted cell apoptosis and decreased drug resistance against cisplatin in some specific cell lines of TSCC (Lv, Yang & Yang, 2020). Chen et al. (2019a) revealed that overexpression of integrin α5 boosted the migration and invasion ability of human mesenchymal stem cell-treated HCC cells. Spinler et al. (2020) demonstrated that the integrin β7 was preferentially expressed in drug-resistant blast crisis chronic myeloid leukemia (bcMCL) stem cells, contributing to the growth and dissemination of bcMCL. Depletion of its upstream syndecan-1 disrupts its function, leading to ideal therapy outcomes. The Sdc1-integrin β7 axis plays a major role in the communication of bcCML and niches. Ramovs et al. (2020) showed that integrin α3β1 in hair bulge stem cells indirectly participated in the formation of a tolerant tumor environment by modulating the expression of matricellular protein connective tissue growth factor (CCN2), which promoted colony formation and transformed keratinocyte growth, contributing to initial skin tumorigenesis.

General role of integrins in CSCs

Other research also identified the general association of integrins and CSCs. Integrin α2β1 is downregulated in carcinomas with a poor differentiation status and concurrently shows an ability to promote metastasis (Ojalill et al., 2018). It was widely confirmed that integrin β1 plays a significant role in promoting the metastasis, chemoresistance, and self-renewal of CSCs (Nisticò et al., 2014). Integrins α6β1, α6β4, and αvβ3 are highly expressed in normal and cancer stem cells and may be involved in the positive regulation of tumorigenesis (Cooper & Giancotti, 2019; Farahani et al., 2014). More specifically, when tumor cells are cultured with collagen, there appears to be a preferential glycosylation-dependent positive selection of CSCs, subsequently triggering their expansion and generation, contributing to enhanced tumorigenic and metastatic potential. Integrin β1 is a mediator of CSC modulation induced by collagen, as knockdown of integrin β1 gene expression and the use of an integrin β1 blocking antibody impaired the interaction between CSCs and the ECM, thus preventing both the initial selection of pre-existing CD133+ CSCs, initially modulated by Glc-collagen, and their subsequent expansion and de novo generation (Gardelli et al., 2021). Moreover, multiple studies have shown that integrin α7 promotes metastasis by inducing the EMT. It has been demonstrated that integrin α7 enhances CSC features, including promoting spheroid formation, cell migration, and invasion through the FAK/MAPK/Erk signaling pathway (Ming et al., 2016). It was also reported that integrin α7 induced FAK/Akt signaling to inhibit apoptosis (Ming et al., 2016). Research demonstrated that, combined with CD90, integrin β3 mediated anti-tumor functions. The overexpression of CD90 suppresses the sphere-forming ability and ALDH activity and enhances cell apoptosis, demonstrating that it may reduce cell growth through CSCs and anoikis. Moreover, CD90, as a CSC marker, can also weaken the expression of other CSC markers, such as CD133 and CD24. However, by replacing the RLD domain of CD90 with the RLE domain, the inhibition of CD133 expression by CD90 was weakened. Significantly, the CD90-mediated inhibition of CD133 expression, anchorage-independent growth, and signal transduction of mTOR and AMPK were restored by integrin β3 shRNA (Chen et al., 2016b).

Integrins are pivotal in modulating CSC characteristics across an array of cancer types. Their pro-tumorigenic role in CSC self-renewal, metastasis, and resistance to therapy underscores their potential as targets for therapeutic intervention. However, current research has not demonstrated direct anti-tumor effects of integrins in their interactions with CSCs, highlighting the need for further investigation. By focusing on specific integrins or their downstream signaling pathways, it may be possible to disrupt CSC maintenance and augment the effectiveness of traditional therapies. Despite this promise, achieving a comprehensive understanding of the complex mechanisms through which integrins regulate CSCs, exploring the diversity of integrin expression in CSCs and its impact on tumor development and therapeutic approaches, and developing targeted therapies accordingly, requires further investigation. Advancing this knowledge has the potential to lead to more personalized and efficacious treatment strategies, ultimately enhancing patient outcomes in the ongoing battle against cancer.

Integrin-related therapy of tumors

Integrins are overexpressed on endothelial cells, CAFs, and CSCs in patients with tumors, and play crucial roles in tumor angiogenesis, progression, metastasis and other processes (Li et al., 2019; Liu et al., 2023b). Consequently, targeted therapy of integrins and their related signaling cascades is imperative to achieve anti-tumor effects with high specificity and few side effects (Ellert-Miklaszewska et al., 2020; Li et al., 2019; Liu et al., 2023b). Targeted therapies focusing on integrins can be categorized into three types based on the processes they affect: direct targeting of integrins, indirect targeting of related integrin signaling components, and integrin-mediated selective drug delivery systems (Table 1).

Table 1 The integrin antagonists and downstream signaling protein inhibitors for tumor therapeutics.

Therapy name	Therapy type	Target; mechanism of action	Clinical trial and current state	Applications	
Cilengitide	RGD peptide	Integrin αvβ3, αvβ5; decreases integrin αvβ3 and αvβ5 expression, and inhibits FAK/Src/Akt pathway, thus promoting apoptosis of tumor cells	Phase III; completed	Glioblastoma, HNSCC, laryngeal cancer	
GLPG0187	Non-peptide	Integrin αv; inhibits TGF-β signaling, thus suppressing breast cancer invasion and metastasis	Phase I; completed	Glioma	
CNTO 95	Monoclonal antibody	Integrin αvβ3, αvβ5; impedes FA and tumor cell motility signal, thereby inhibiting breast tumor metastasis	Phase I; completed	Advanced refractory solid tumors	
Abegrin	Monoclonal antibody	Integrin αvβ3, αvβ5; promote endothelial and tumor cells apoptosis, thus inhibiting melanoma, lymphoma, and gastric cancer progression	Phase II; completed	Melanoma, lymphoma, gastric cancer	
Vitaxin	Monoclonal antibody	Integrin αvβ3, αvβ5; suppresses vitronectin and osteoclast adhesion, and angiogenesis, thus impairing tumor progression and bone resorption	Phase II; completed	Melanoma, lymphoma, gastric cancer	
Selective A2 aptamer	DNA aptamer	Integrin β1; acts as DNA nano-carrier transporting doxorubicin, thereby presenting anti-tumor effect	In vitro and in vivo study	Esophageal squamous cell carcinoma	
D-pinitol		Integrin αvβ3; impairs FAK/c-Src and NF-κB signaling, thereby inhibiting tumor invasion and metastasis	In vitro study	Breast cancer	
Defactinib	FAK inhibitor	FAK-Y925, FAK-Y397; inactivates FAK downstream PI3K/Akt/STAT3 pathway, thus inhibiting pancreatic ductal adenocarcinoma progression	Phase II; ongoing	Pancreatic ductal adenocarcinoma, merlin-low malignant pleural mesothelioma	
Idelalisib	PI3K inhibitor	PI3Kγ; combines with ofatumumab, thus ameliorating chronic lymphocytic leukemia	Phase III; terminated	Chronic lymphocytic leukemia	
Dasatinib	Src inhibitor	Src; inhibits Src and downstream signaling, thereby suppressing chronic myeloid leukemia	Phase III; completed	Chronic myeloid leukemia, Philadelphia chromosome-positive acute lymphoblastic leukemia	
Bosutinib	Src inhibitor	Src; inhibits Src and downstream signaling, thereby suppressing chronic myeloid leukemia	Phase III; completed	Chronic myeloid leukemia	
Oxymatrine		Integrin αvβ3; impedes integrin αvβ3-mediated FAK/PI3K/Akt signaling, thus hindering breast cancer metastasis	In vitro study	Breast cancer	
ATN-161	Pentapeptide	Integrin α5β1; interferes binding to fibronectin, thus impeding prostate cancer angiogenesis, progression and metastasis	Phase II; completed	Prostate cancer, breast cancer, solid tumors	
Volociximab	Monoclonal antibody	Integrin α5β1; interferes binding to fibronectin, thus impeding prostate cancer angiogenesis, progression and metastasis	Phase II; completed	Ovarian cancer, peritoneal cancer	
Tinagl1	Antibody	Integrin α5β1, αvβ1; impairs FAK pathway, thus repressing triple negative breast cancer progression and metastasis	In vitro and in vivo study	Triple-negative breast cancer	
Gleditsia sinensis		Integrin α2β1; decreases integrin α2β1 expression, thus inhibiting lung and breast cancer progression	In vitro study	Lung cancer, breast cancer, prostate cancer	
Alternagin-C	Disintegrin protein	Integrin α2β1, β1, VEGFR2; promotes metastasis suppressor 1 expression, suppresses MMP9/2 expression, and inactivates Erk1/2 /PI3K and FAK/Src pathways, thus repressing cancer metastasis	In vitro study	Triple-negative breast cancer	
Tanshinone IIA	Targeted therapy	Integrin β1 mRNA, MMP-7 mRNA; decreases integrin β1 and MMP-7 expression, thereby interfering gastric cancer metastasis	In vitro study	Gastric cancer	
Chrysotobibenzyl	Targeted therapy	Integrin β1; suppresses FAK, and Akt pathway, thereby inhibiting lung cancer progression	In vitro study	Lung cancer	
Curcumin		Integrin β1; decreases integrin β1 expression, thereby inhibiting cancer progression	Phase II; completed	Pancreatic cancer, colon cancer	
Notes.

Abbreviations FA focal adhesion

FAK focal adhesion kinase

Src Src family kinase

PI3K phosphatidylinositol 3-phosphokinase

Erk extracellular regulated protein kinase

Akt protein kinase B

STAT3 signal transducer and activator of transcription 3

VEGFR2 vascular endothelial growth factor receptor 2

MMP matrix metalloproteinases

TGF-β transforming growth factor β

NF-κB nuclear factor kappa B

HNSCC head and neck squamous cell carcinoma

Targeted therapy

Targeting specific integrins

Various drugs that directly target integrin αvβ3 and αvβ5 represent different modalities, including RGD peptides, non-peptides, monoclonal antibodies, and nucleic acid aptamers (Ellert-Miklaszewska et al., 2020; Li et al., 2019; Liu et al., 2023b; Zhang et al., 2022b). Cilengitide, a cyclic RGD peptide integrin inhibitor with great safety and tolerance, can decrease the expression of integrin αvβ3 and αvβ5 and inhibit the FAK/Src/Akt signaling pathway, thus triggering apoptosis in glioblastoma, HNSCC and laryngeal cancer cells (Ellert-Miklaszewska et al., 2020; Li et al., 2019; Liu et al., 2023b; Stupp et al., 2014). In addition, conjugating with radiotherapy, Cilengitide poses the potential for the utilization of this combined therapy for glioblastoma patients with O6-methylguanine DNA methyltransferase (MGMT) methylation in a completed 5-year analysis of the European Organization for Research and Treatment of Cancer (EORTC) and National Cancer Institute of Canada (NCIC) trial (Robinson & Hodivala-Dilke, 2011). However, increasing studies pointed out that Cilengitide at low concentrations promotes tumor angiogenesis rather than inhibiting it, which is due to low-dose Cilengitide can activate Rab-4-mediaed VEGFR2 recycling pathway. Further, steady-state VEGFR2 is translocated from intracellular compartment to endothelial cell surface, facilitating VEGF pro-angiogenic effects (Bazzazi et al., 2018). A phase III CENTRIC trial and a phase II CORE trial both in completed state and investigated that Cilengitide failed to improve progression-free survival (PFS) and overall survival (OS) for glioblastoma patients without MGMT methylation, which precludes its clinical utilization (Li et al., 2019; Liu et al., 2023b). Apart from dosing, timing of prescribing Cilengitide also affect its clinical efficacy. In in vivo study, Steri et al. (2014) revealed that gene ablation of integrin αvβ3 manipulated before tumor growth rather than tumor establishment displayed beneficial anti-angiogenic effects in melanoma and lung cancer cell lines. Among non-peptide antagonists, GLPG0187 has emerged as a novel therapy impeding glioma invasion and metastasis by inhibiting integrin αv and TGF-β in a completed phase I dose escalating study (Cirkel et al., 2016). In phase I clinical trials of advanced solid tumors, GLPG0187 exhibited acceptable safety and good tolerance (Ellert-Miklaszewska et al., 2020; Li et al., 2019). Monoclonal antibodies such as CNTO 95 (intetumumab), 17E6, Abegrin (MEDI-522 or etaracizumab) and Vitaxin (MEDI-523) bind to integrin αvβ3 and αvβ5 with varying affinities. For example, CNTO 95 interferes with integrin αv involved in focal adhesions and cell motility signals in vitro in breast tumor cells (Chen, Zhao & Xie, 2022; Li et al., 2019). Combined with radiotherapy, it exhibited anti-angiogenesis function in mice with various human cancer xenografts (Ning et al., 2008). Moreover, Jia et al. (2013) demonstrated that the combination of CNTO 95 and dasatinib dually impeded integrin αv and Src in vitro showing potent anti-angiogenesis effects in human umbilical vein endothelial cells. Regarding its clinical results, Mullamitha et al., (2007) showed that CNTO 95 was safe and well-tolerated in patients with refractory solid tumors and its dose-dependent mean half-life ranged from 0.26 to 6.7 days in a completed phase I clinical trial. Nevertheless, Heidenreich et al. (2013) conducted a randomized, double-blind, phase II clinical trial for patients with metastatic castration-resistant prostate cancer and highlighted that conjugating CNTO 95 with docetaxel and prednisone failed to improve PFS, OS, and prostate-specific antigen (PSA) response than placebo group. Vitaxin exhibits anti-angiogenic effects by promoting the apoptosis of endothelial cells in newly formed blood vessels, thus inhibiting tumor nourishment and progression and impairs vitronectin and osteoclast adhesion to suppress bone resorption in completed phase II studies (Chen, Zhao & Xie, 2022). Phase I and II clinical trials of Vitaxin both showed the cancer stabilization, acceptable safety, good tolerance, and no serious toxicity (Borst et al., 2017; Gutheil et al., 2000). Nevertheless, a pilot phase I study did not display improvement in tumor angiogenesis and regression for patients with metastatic cancer receiving intravenous doses of 10, 50 or 200 mg Vitaxin (Posey et al., 2001). Abegrin induces endothelial and tumor cell apoptosis in melanoma, lymphoma, and gastric cancer, displaying a higher affinity for integrin αvβ3 than Vitaxin in a randomized phase II study (Chen, Zhao & Xie, 2022; Hersey et al., 2010; Li et al., 2019). Conjugating Abegrin with labelling and other substances can be applied in radioimmunotherapy, cancer monitoring, and drug dose optimization. Completed phase I clinical trials showed that Abegrin is well-tolerated without significant toxicity (Chen, Zhao & Xie, 2022; Veeravagu et al., 2008). However, Hersey et al. (2010) found that either Abegrin alone or conjugated with dacarbazine failed to meet primary endpoints in randomized phase II clinical research for stage IV metastatic melanoma. Apart from these clinical medicines, several preclinical integrin αvβ3 antagonists show promising anti-tumor effects, including benzyl guanidine-PEG-triazole tetraiodothyroacetic acid (BG-P-TAT), ST1646, fb-PMT, and obtustatin (Liu et al., 2023b). DNA and RNA aptamers, characterized by their three-dimensional structure and great affinity and specificity for integrin αvβ3, can impede tumor angiogenesis and progression in in vitro study (Das et al., 2018; Zhang et al., 2022b). Moreover, Zhang et al. (2022b) revealed that the selective A2 aptamer bound to integrin β1 via a shared RGD motif and acted as a DNA nano-carrier with doxorubicin (Dox), thereby exhibiting an anti-tumor effect in vitro and in vivo at target sites in esophageal squamous cell carcinoma. Additionally, several natural products also can inhibit integrin αvβ3 function. For example, D-pinitol, a chemical derived from plants, can hinder αvβ3 expression, FAK/c-Src kinase phosphorylation, and p65 phosphorylation in the NF-κB signaling pathway, thus inhibiting prostate cancer invasion and metastasis in vitro (Li et al., 2019).

In addition to integrin αvβ3 and αvβ5, peptides and monoclonal antibodies that directly bind to other integrins are also under investigation. First, by targeting integrin α5β1, ATN-161, a pentapeptide, interferes with FN adhesion to integrin α5β1, thereby repressing prostate cancer and breast cancer angiogenesis, progression, and metastasis in vitro and in vivo (Chen, Zhao & Xie, 2022; Li et al., 2019). Cianfrocca et al. (2006) showed that ATN-161 was a well-tolerated, anti-angiogenesis, and anti-tumor drug exhibiting disease stabilization in a phase I clinical study. However, ATN-161 only displayed tumor stabilization, rather than achieving primary anti-tumor effect in patients with solid tumors. A phase II clinical trial treating recurrent malignant glioma patients with ATN-161 and carboplatin is completed. Nonetheless, Khalili et al. (2006) found that ATN-161 only exerted anti-breast cancer proliferation in vivo rather than in vitro. Volociximab, a monoclonal antibody with high affinity, exhibits pharmacological effects similar to those of ATN-161 (Das et al., 2018; Ellert-Miklaszewska et al., 2020; Li et al., 2019). Both completed phase I and II clinical trials demonstrated that Volociximab was confirmed to be an effective and well-tolerated drug without significant side effects (Bell-McGuinn et al., 2011; Chen, Zhao & Xie, 2022). However, Bell-McGuinn et al. (2011) conducted single-arm phase II research which highlighted the insufficient clinical efficacy of Volociximab in patients with refractory epithelial ovarian or primary peritoneal cancer. Tubulointerstitial nephritis antigen-like 1 (Tinagl1) directly targets integrin α5β1 and αvβ1, subsequently suppressing the FAK pathway and impeding TNBC progression and metastasis in vitro and in vivo (Shen et al., 2019). Second, by targeting integrin α2β1, Gleditsia sinensis from bean agaric downregulates integrin α2β1 expression and inactivates integrin α2β1/FAK/Src signaling pathway in lung, breast, and prostate cancer in vitro (Li et al., 2019). Alternagin-C (ALT-C), a disintegrin protein from venom, binds to integrin α2β1 and promotes metastasis suppressor 1 (MTSS1) expression, thereby reducing MMP-9/2 expression and metastasis in the MDA-MB-231 triple-negative breast cancer cell line in vitro (Dos Santos et al., 2020; Moritz et al., 2022). Additionally, Dos Santos et al. (2020) demonstrated that ALT-C can also dephosphorylated VEGFR2 and the integrin β1 subunit, and inhibited Erk1/2 /PI3K and FAK/Src signaling pathways in vitro, which further represses tumor angiogenesis. Third, by targeting integrin β1, curcumin can decrease its expression, whose efficacy and safety in advanced pancreatic cancer are studied in a completed phase II clinical trials (Li et al., 2019). Apart from decreasing the expression of integrin β1, amygdalin causes an extra reduction in MMP-2/9 in lung cancer and suspends bladder cancer cells in the S phase or G0 /G1 phase, thus impeding these cancer cell proliferation, migration, and invasion in vitro (Li et al., 2019). Additionally, tanshinone IIA suppresses gastric cancer metastasis by reducing integrin β1 and MMP-7 mRNA transcription in vitro (Chen, Zhao & Xie, 2022; Li et al., 2019). However, dual targeting of integrin β1 and heparan sulfate exhibits greater metastasis inhibition in pancreatic cancer than either approach alone (Roy et al., 2020). Petpiroon et al. (2019) showed that chrysotobibenzyl inhibited β1 expression in both H460 and H292 lung cancer cell lines, hindered FAK and Akt signaling pathways, and suppressed EMT in vitro, which eventually impeding lung cancer metastasis. Furthermore, inhibiting integrin β1 and activating FAK signaling by the antibody-drug conjugate ABBV-085, which targets LRRC15, represses ovarian cancer dissemination (Ray et al., 2022).

Targeting the downstream signaling pathway of integrin

Instead of directly targeting the integrins themselves, we focused on inhibitors of key signaling molecules involved in integrin signaling pathways, specifically targeting FAK, PI3K, Src, and Rac. In clinical applications, as a FAK inhibitor, defactinib can dephosphorylate FAK-Y925 and FAK-Y397 by competing with adenosine triphosphate (ATP) in clinical trials related to PDAC (Liu et al., 2023b). Defactinib was proven to exhibit great safety and well-tolerance in phase I clinical trials as a monotherapy. In addition, Jiang et al. found that VS-4718, a FAK inhibitor, can facilitate the infiltration of CD8+ CTL, decrease the numbers of immunosuppressive cells, such as tumor-infiltrating myeloid-derived suppressor cells, CD206+ tumor-associated macrophages, and CD4+FOXP3+ Tregs, and improve the response to anti-PD1 and gemcitabine in the p48-Cre/LSL-KrasG12D/p53Flox/+ (KPC) PDAC mouse model (Jiang et al., 2016). Furthermore, Wang-Gillam et al. (2022) showed that the combination of defactinib, pembrolizumab, an anti-PD1 mono-antibody, and gemcitabine was well-tolerated and safe in phase I, dose escalation, and expansion study. There is phase II clinical research in recruiting state investigating pembrolizumab with or without defactinib as a neoadjuvant and adjuvant therapy for resectable PDAC. However, Fennell et al. (2019) indicated that there was no significant improvement in PFS and OS of patients with merlin-low malignant pleural mesothelioma in a phase II clinical trial, when defactinib was used as a maintenance therapy. GSK2256098, another ATP-competitive FAK inhibitor, impede the FAK downstream pathway, PI3K/Akt/STAT3, to inhibit HepG2 progression in a preclinical study (Liu et al., 2023b). Phase I clinical trial of GSK2256098 showed that it was effective, safe, and well-tolerated in advanced solid tumors including merlin-loss mesothelioma and recurrent glioblastoma (Brown et al., 2018; Liu et al., 2023b; Soria et al., 2016). Brastianos et al. (2023) showed that GSK2256098 was well-tolerated, had acceptable safety profile, and improved the PFS of patients with meningiomas in a phase II clinical trial. Alpelisib and idelalisib are PI3Kα and PI3Kγ inhibitors, respectively, and represent remarkable clinical improvement when combined with other drugs. For example, combining idelalisib with the CD20 inhibitor ofatumumab improved PFS by two-fold among chronic lymphocytic leukemia patients in a terminated, open-label, and randomized phase III study (Jones et al., 2017). Moreover, phase III clinical trials in completed state revealed that the combination of alpelisib and fulvestrant improved the PFS and median overall survival in patients with PIK3CA mutation, hormone receptor-positive, and human epidermal growth factor receptor-negative advanced breast cancer (André et al., 2019; André et al., 2021). Additionally, dasatinib and bosutinib are Src inhibitors used to treat chronic myeloid leukemia (CML) patients (Liu et al., 2023b; Porkka et al., 2010). Phase II clinical trials in completed state showed that dasatinib was well-tolerated and triggered significant molecular response in patients with Philadelphia chromosome-positive acute lymphoblastic leukemia (Foà et al., 2011). Cortes et al. (2016) indicated that dasatinib 100 mg once daily was the first-line therapy for the long-term treatment of CML patients in chronic phase exhibiting great safety and efficacy in a completed, open-label, and DASISION phase III clinical study. Regarding to the clinical results of bosutinib, a phase II clinical trial in completed state showed that it had acceptable safety profile and three most common adverse events including diarrhea, increased alanine aminotransferase, and increased aspartate aminotransferase in Japanese patients with CML in chronic phase (Hino et al., 2020). Cortes et al. (2017) revealed that in an open-label phase I/II clinical research long term use of bosutinib in patients with philadelphia chromosome-positive leukemias could induce reversible estimated glomerular filtration rate decline. Furthermore, compared with asciminib, bosutinib exhibited lower major molecular response rate, fewer adverse events, and treatment discontinuation in a completed phase III clinical study (Réa et al., 2021). Notably, Ruegg, Dormond & Mariotti (2004) reported that novel drugs targeting integrin αvβ3-mediated Rac activation can be designed to suppress Rac activation-induced endothelial cell migration, proliferation and angiogenesis. Moreover, lipoic acid involvement in endothelial cell EGFR signaling activates AMPK and inhibits the Akt and mTOR pathways, suppressing tumor cell proliferation. In addition, LA-induced ROS generation decreases anti-apoptotic Bcl2 and upregulates pro-apoptotic proteins, promoting tumor cell apoptosis in vitro (Puchsaka, Chaotham & Chanvorachote, 2016). Furthermore, cellular superoxide anion (O2−) and hydrogen peroxide (H2O2) generated by LA cause chemotherapeutic sensitization and inhibit lung cancer metastasis via inhibiting integrin β1/3 expression in vitro (Farhat & Lincet, 2020). In completed phase I and II trials, LA was confirmed to reduce the paclitaxel- and doxorubicin-induced peripheral neuropathy in patients with breast cancer (Melli et al., 2008; Werida et al., 2022). Oxymatrine in breast cancer cells suppresses integrin αvβ3/FAK/PI3K/Akt signaling to reduce metastasis in vitro (Chen et al., 2019b).

Targeted therapy using an integrin-mediated selective drug delivery system

In integrin-mediated drug delivery systems, RGD peptides or peptidomimetics that bind to specific integrins on the surface of tumors and other abnormal cells are incorporated into nano-carriers, nano-assemblies, liposomes or exosomes containing drugs or radionuclides for targeted therapy and diagnosis (Das et al., 2018; Egorova & Nikitin, 2022; Ellert-Miklaszewska et al., 2020; Fu et al., 2021; Moasses Ghafary et al., 2022). For example, RGD-based carriers deliver DNA or siRNA to modulate gene expression and transport cytotoxic anti-tumor drugs such as Dox (Chauhan et al., 2021; Das et al., 2018; Ellert-Miklaszewska et al., 2020; Moasses Ghafary et al., 2022). As mentioned above, abergin, a monoclonal antibody targeting to integrin αvβ3, participates in radio-immunotherapy and imaging in glioblastoma (GBM) with high precision and specificity when labeled with 60Y (Ellert-Miklaszewska et al., 2020). Moreover, PEG-PLA with an RGD motif loads paclitaxel and docetaxel, addressing their low permeability through the blood–brain barrier (BBB) and achieving targeting effects. Notably, the cyclic peptide iso-DGR emerged recently as a novel integrin αvβ3-binding motif similar to the RGD motif, albeit without activating the integrin αvβ3 (Pang et al., 2023). Therefore, to some extent, it is regarded as an integrin antagonist. Similarly, polymeric micelles (PMs) serve as nanoparticles (NPs) to deliver drugs. Chauhan et al. (2021) indicated that in GBM, the cyclic peptide Arg-Gly-Asp-Phe-Val on the surface of PM assisted in transporting pitavastatin to endothelial cells in the BBB and to tumor cells, exhibiting precise tumor localization via SiO2 on the surface of the PMs, as shown using fluorescence microscopy and flow cytometry. Likewise, polymersomes targeting integrin α3 loaded with volasertib, a polo-like kinase 1 (PLK1) inhibitor, displayed remarkable internalization of the drug in SKOV-3 ovarian cancer cells (Wang et al., 2021). Currently, modified exosomes are also considered novel drug delivery systems for solid tumors (Shao, Zaro & Shen, 2020). For example, HEK-293T cells modified with the integrin αv-specific iRGD peptide and carrying Dox treat anaplastic thyroid carcinoma. In addition, labeling with 131I can aid in vivo imaging via single-photon emission computed tomography (CT) (Wang et al., 2022). Notably, targeting technology contributes to both targeted chemotherapy and radiotherapy. C(RGDyC)-AuNPs, 64Cu-Pyro-3PRGD2, and 3PRGD2 all target integrin αvβ3 with different affinities, improving the radiosensitivity and efficacy of radiotherapy in positron emission tomography (PET) (Yu et al., 2023).

To conclude, integrin-targeted therapies utilize strategies like RGD peptides, antibodies, and aptamers against αvβ3/αvβ5 integrins, though clinical challenges persist. Cilengitide’s failure underscored angiogenesis regulation complexity, while agents such as GLPG0187 and Abegrin yielded inconsistent outcomes. Research extends to downstream signaling inhibitors (defactinib for FAK, alpelisib for PI3K) often combined with conventional treatments. RGD-based nanocarriers improve tumor-specific drug delivery, yet limitations remain, including high trial attrition rates, toxicity, and resistance. Emerging approaches emphasize rational combination therapies, next-generation inhibitors with enhanced selectivity, and biomarker-guided personalized strategies. Advanced nanotechnology explores pH-sensitive or dual-targeting delivery systems, while mechanistic studies focus on resistance pathways like integrin recycling and compensatory signaling. Tumor microenvironment modulation and structural biology advances (cryo-EM mapping) aid in developing isoform-specific inhibitors. Multidisciplinary efforts integrating nanotechnology, immunotherapy, and AI-driven design aim to overcome therapeutic barriers, with CRISPR screening further clarifying resistance mechanisms for targeted interventions.

Conclusions

As cell adhesion receptors and mechanoreceptors on the plasma membrane, integrins regulate intercellular and cell-matrix crosstalk through outside-in and inside-out signaling pathways. Extensive studies have shown that integrins regulate endothelial cell viability and proliferation, thus triggering tumor angiogenesis, lymph angiogenesis, and metastasis (McKay et al., 2020; Sokeland & Schumacher, 2019). On CAFs, integrins contribute to tumor progression and invasion in response to mechanical stress (Brown & Marshall, 2019; Henderson, Rieder & Wynn, 2020; Jang & Beningo, 2019). It was demonstrated that integrins also participate in the modulation of stemness, chemoresistance, and survival of CSCs, thus contributing to cancer initiation and progression. Moreover, several studies have also revealed that dysregulated expression of integrins affected the evasion of immune response and immune tolerance, further promoting tumor overgrowth (Zhang et al., 2023). Therefore, abnormal integrin expressions, mutated genes related to integrin, and aberrant downstream or upstream signaling pathways can potentiate tumor development.

Current targeted therapies primarily focus on three types: direct targeting of specific integrins, indirect targeting of signaling pathways induced by integrin activation, and novel integrin-based drug delivery systems. These therapies offer high specificity and good tolerance and hold promise for improving the prognosis and efficacy of conventional therapeutics. Additionally, integrins are used in targeted diagnosis to enhance accuracy and efficiency. For example, multiple modified small-molecule integrin antagonists are used as imaging tools, such as [18F]FBA-A20FMDV2 and [18F]FP-R01-MG-F2, enabling precise detection of diseases such as idiopathic pulmonary fibrosis in PET examinations in vivo (Slack et al., 2022). Furthermore, NPs modified with integrin αvβ3, such as C(RGDyC)-AuNPs, can optimize the sensitivity of CT imaging in vivo and the efficacy of radiotherapy of tumors (Yu et al., 2023).

Despite translational potential, integrin-targeted therapies face clinical challenges including pleiotropic signaling cascade-induced off-target effects, spatiotemporal heterogeneity in isoform expression across tumor microdomains, and lack of validated stratification biomarkers. Hypoxia-driven integrin conformational switching and mechano-transductive feedback loops in the tumor microenvironment perpetuate therapeutic resistance through dynamic target modulation. Critical knowledge gaps persist in integrin-mediated cancer stem cell niche maintenance and exosomal communication networks. Advanced models, such as orthotopic patient-derived xenografts, and CRISPR-engineered murine platforms with humanized stroma enhance preclinical predictive validity. Nano-diagnostics and nano-therapeutics based on integrin conformation improve tumor-selective delivery via ligand affinity, while rational polytherapy combining integrin αvβ3 antagonists with checkpoint inhibitors exploits synthetic lethality. Standardized multi-parametric profiling integrating phosphor-proteomics and biomechanical mapping remains imperative. Innovative strategies prioritize microenvironment-activated prodrugs and bispecific engagers targeting integrin/co-receptor complexes. Clinical translation requires tissue-selective delivery systems, AI-driven biomarker discovery, and multidimensional biomarker algorithms leveraging single-cell interactome mapping. Machine learning-driven deconvolution of stromal-integrin crosstalk networks and spatiotemporal resolution of mechanochemical signaling emerge as strategic priorities. Overcoming these barriers could establish integrin modulation as a cornerstone of precision stroma-oncology, transforming therapeutic paradigms in epithelial-mesenchymal transition-driven malignancies.

We thank LetPub for its linguistic assistance during the preparation of this manuscript.

Additional Information and Declarations

Competing Interests

Author Contributions

Data Availability

The authors declare there are no competing interests.

Yifan Li conceived and designed the experiments, performed the experiments, analyzed the data, prepared figures and/or tables, authored or reviewed drafts of the article, and approved the final draft.

Shantong Peng conceived and designed the experiments, performed the experiments, analyzed the data, prepared figures and/or tables, authored or reviewed drafts of the article, and approved the final draft.

Jiatong Xu analyzed the data, authored or reviewed drafts of the article, and approved the final draft.

Wenjie Liu analyzed the data, authored or reviewed drafts of the article, and approved the final draft.

Qi Luo analyzed the data, authored or reviewed drafts of the article, and approved the final draft.

The following information was supplied regarding data availability:

This is a literature review.

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
