# Peer review of "Integrin signaling in tumor biology: mechanisms of intercellular crosstalk and emerging targeted therapies"

_PeerJ, doi:10.7717/peerj.19328_

## Round 0.1 · original submission · Major Revisions

The three reviewers agree that your paper will contribute to our understanding of integrin-targeted cancer therapy. However, they have pointed out a number of areas that need revision. Please revise your manuscript according to the reviewers' comments. In particular, revise the Discussion section according to the comments of Reviewer 2 and use appropriate wording choices according to Reviewer 3's suggestions.

Reviewer 1 ·

Basic reporting

This is a thorough and generally well written review that covers an interesting area that has not been reviewed recently, focused on integrin receptors and targeting them in cancer therapeutics. The review is of broad interest to a range of fields with a good introduction and rationale.

Experimental design

The methodology for identifying papers used in this review is clear and unbiased with a good number of citations. Overall, the review is well structured and flows nicely.

Some of the citations are described as "another scientist..." (e.g. line 297). This sounds a bit vague and names should be used, as they have for other papers throughout, or this type of description removed entirely.

I would recommend breaking up the CSC section somehow. It is currently a very long section and is difficult to read, it is of considerable interest though.

Validity of the findings

Whilst the overall argument and support are described well, each of the sections could benefit from a concluding statement that highlights what has been described about integrins in each context and what open research questions there are. It currently reads more like a list of findings rather than offering any opinion on unresolved questions, which in itself is interesting and useful but some further discussion would elevate the piece somewhat. The current conclusion is quite general, whereas some specific comments would help greatly.

I would also recommend to the authors that they change the title of the piece. They do not actually discuss integrins in cell-cell interactions, rather just describe the role of integrins in different cell types and I think that this is currently a little misleading. Similarly, the main section titled “effects of integrins on tumour progression” doesn’t really discuss this directly. Rather it describes potential roles for integrins in some key processes associated with tumour biology, so I would recommend changing this as well.

Additional comments

I have several minor comments and changes that require addressing:

Example on line 78 doesn’t make sense. Sentence before says that integrins can interact with other cell surface proteins which contribute to signalling, so the subsequent example should be something to demonstrate this point e.g. growth factor receptors.

Line 106 needs clarification as “fields of integrin and tumor” doesn’t make sense. Perhaps “integrins and cancer research” or “integrins and tumour biology”? Similarly, I wouldn’t describe integrins as communicating with endothelial cells I would change it to something like “ function of integrins in tumour progression via signalling in endothelial cells…”

Delete word “promisingly” on line 110

Line 114 needs rewriting, “but there is less research fully explained them” could be changed to “but there is little research fully investigating them”.

Line 186, instead of “besides” say “in addition”

Line 254, I was under the impression that extravasation was when cells exited the tumour by infiltrating blood vessels. I think what has been described here are the initial stages on intravasation that are more associated with endothelial cell attachment in a similar manner to immune cells.

Line 321, “proteins” rather than “substances”?

Line 415, “Laminin” instead of “adminin”?

Line 615, the opening sentence of this section requires some rewriting. I would recommend discussing each FAK inhibitor separately. In addition, I would recommend adding some brief details about the other therapeutics mentioned that are used in conjunction with the specific FAK inhibitors.

Figure 1 has typos in “intracellular matrix” and “laminin”

·

Basic reporting

This literature review by Li et al., focuses on integrin-mediated crosstalk between the tumor and the tumor microenvironment, and on current integrin-targeting therapeutics. The topic is timely, however, the following items should be addressed in preparation for publication.

1. Overall, this review presents a very myopic viewpoint as all integrins/integrin-mediated signaling as being pro-tumorigenic. The reality is that some integrins/ integrin-mediated signaling have been shown to be anti-tumorigenic in some contexts. Whether an integrin is pro- or anti-tumorigenic can be cancer type, subtype, or even stage specific. The complexity is enhanced further when the same integrin can be pro- or anti-tumorigenic depending on which cell type expresses it. This complexity is, in part, why integrin targeting therapeutics have been unsuccessful overall. To that point, the review presents the integrin-based therapeutics rather optimistically, while in reality they have been less successful. The authors should take care to present a more accurate and complete picture of the roles of integrins in the tumor setting.

2. Much of the information throughout, and particularly in the tumor angiogenesis section is unclear as written. For example, often times it is unclear whether the integrin being discussed is present on the tumor cell, or the stromal cell (e.g., endothelial cell, lymphatic cell, CAF). It is also unclear which context the integrin was studied (e.g., which cancer(s) types, or physiological condition? In vitro studies, or preclinical animal model?)

3. The Cancer Stem Cells section needs to be re-organized. There is a lot of information, but it is not presented in a logical way. Perhaps organizing by integrin or by cancer type would be helpful to distinguish the studies presented. The sentence on lines 403-404 is unclear as written. Line 424, how can integrin relocate if it’s knocked out? Please clarify.

4. It should be clarified in figures 2-4 that these are summarizing pro-tumorigenic integrin-mediated processes that are context-dependent (not occurring on all tumor cells).

5. The term “cell-cell interaction” can be misinterpreted as direct cell-cell adhesion (e.g., via cadherins or something of the sort). I believe the authors instead are referring to intercellular crosstalk that is mediated by integrins through secreted factors. This needs to be clarified throughout, and the term “cell-cell interaction” should be replaced in the manuscript title and throughout the text.

6. In the targeted therapy section, it should be clarified if the studies presented are in vitro results, or preclinical animal models or from the clinic (e.g., Abegrin findings on line 568-570). Further, it should be noted in Table 1 and in the text, which, if any, are currently used in the cancer clinic or which failed during clinical trial, or which are still ongoing.

7. Table 1. If no clinical trial phase is indicated (e.g. for Selective A2 aptamer, D-pinitol, etc.), does this mean it is in pre-clinical phase or clinically in use? Either way, this should be clearly indicated in the table.

8. Conclusion. The discussion on the challenges of integrin targeting should be expanded.

Experimental design

No comment. (This is a literature review.)

Validity of the findings

No comment. (This is a literature review.)

Additional comments

(Minor)

1. Please check for typos and grammar throughout (e.g., Figure 1. Laminin is misspelled in 2 spots; Line 415 laminin is misspelled).

2. Ensure all information is appropriately referenced (e.g., missing reference on line 95)

3. It is mentioned that there are five groups of integrins (line 64), while they are traditionally categorized into four groups.

4. Is the last paragraph (lines 731-733) supposed to be included? Unclear as written.

Reviewer 3 ·

Basic reporting

This manuscript reviews the role of integrins on tumor progression and considers the potential of integrin-related targets as cancer therapies. The review is quite comprehensive and considers integrin signaling in tumor progression in great depth. The authors are to be congratulated for the comprehensive nature of the review. However, there are several points to consider. Some of these issues seem to arise from the dependence on a word selection program that does not necessarily choose the best wording in the appropriate scientific context.
1. It is inappropriate to refer to integrins as a ‘superfamily” (line 38) which has a precise genetic definition. Integrins are a “family” of transmembrane proteins, albeit a larger family.
2. What is the purpose of the last paragraph of the discussion, 731-733? Are these instructions from the editor.
3. Some of the examples of less than optimal word selections are:
l. 40: 24 alpha and 8 beta subunit combinations,
l.57: “comprise” better “composed of”
l. 63: “author” target audience?
l. 109: “the corresponding integrin-mediated targeted therapy and selective drug delivery system promisingly”?
1.114: “fully explained”
l.127:” literatures”- publications
l. 131: “are responsible’ – regulate
l. 215 “correlate” what does this mean
l. 487. ‘Some specific CSCs have been less reported”?

Experimental design

this is a literature review

Validity of the findings

Comprehensive

---

## Round 0.2 · accepted · Accept

The reviewers confirm that the authors have addressed most of the comments raised by the reviewers. Therefore, I am happy to accept this manuscript for publication in PeerJ.

A minor revision was required by one of the reviewers. Please revise the manuscript upon galley proof.

Reviewer 1 ·

Basic reporting

No comment

Experimental design

No comment

Validity of the findings

No comment

Additional comments

The authors have addressed all of my comments on the original manuscript and I thank them for taking the time to do this to a high standard.

·

Basic reporting

No comment.

Experimental design

No comment.

Validity of the findings

No comment.

Additional comments

The authors have been responsive to the reviewer comments and the amended manuscript is better organized, clarified, and much improved overall.

(Very) Minor revision required: Please proof for grammar and typos throughout (e.g., Fig.1- Intracellular Matrix is spelled “Martix”).